# Adaptive GCN and Bi-GRU-Based Dual Branch for Motor Imagery EEG Decoding

**DOI:** 10.3390/s25041147

**Published:** 2025-02-13

**Authors:** Yelan Wu, Pugang Cao, Meng Xu, Yue Zhang, Xiaoqin Lian, Chongchong Yu

**Affiliations:** School of Computer and Artificial Intelligence, Beijing Technology and Business University, Beijing 100048, China; caopugang0121@163.com (P.C.); 2330602077@st.btbu.edu.cn (M.X.); yy18201485628@163.com (Y.Z.); lianxq@263.net (X.L.); chongzhy@vip.sina.com (C.Y.)

**Keywords:** brain–computer interface (BCI), motor imagery (MI), electroencephalography (EEG), adaptive graph convolutional network (Adaptive GCN), attention module, channel correlation, temporal dependence

## Abstract

Decoding motor imagery electroencephalography (MI-EEG) signals presents significant challenges due to the difficulty in capturing the complex functional connectivity between channels and the temporal dependencies of EEG signals across different periods. These challenges are exacerbated by the low spatial resolution and high signal redundancy inherent in EEG signals, which traditional linear models struggle to address. To overcome these issues, we propose a novel dual-branch framework that integrates an adaptive graph convolutional network (Adaptive GCN) and bidirectional gated recurrent units (Bi-GRUs) to enhance the decoding performance of MI-EEG signals by effectively modeling both channel correlations and temporal dependencies. The Chebyshev Type II filter decomposes the signal into multiple sub-bands giving the model frequency domain insights. The Adaptive GCN, specifically designed for the MI-EEG context, captures functional connectivity between channels more effectively than conventional GCN models, enabling accurate spatial–spectral feature extraction. Furthermore, combining Bi-GRU and Multi-Head Attention (MHA) captures the temporal dependencies across different time segments to extract deep time–spectral features. Finally, feature fusion is performed to generate the final prediction results. Experimental results demonstrate that our method achieves an average classification accuracy of 80.38% on the BCI-IV Dataset 2a and 87.49% on the BCI-I Dataset 3a, outperforming other state-of-the-art decoding approaches. This approach lays the foundation for future exploration of personalized and adaptive brain–computer interface (BCI) systems.

## 1. Introduction

Brain–computer interface (BCI) technology, as a cutting-edge interdisciplinary field, aims to achieve direct communication between the brain and external devices [1,2], bringing unprecedented possibilities to numerous fields such as medicine [3], rehabilitation [4], and smart homes. Within this realm, Motor Imagery–BCI (MI–BCI) is the most commonly used paradigm. Leveraging the brain’s ability to generate neural signatures during mental motor simulation, electroencephalography (EEG) is commonly used to record these correlates non-invasively. When engaged in motor imagery, the sensorimotor cortex activates similarly to physical movement, inducing characteristic scalp electrical potential fluctuations recorded as EEG signals, especially within the alpha and beta bands where event-related desynchronization (ERD) and event-related synchronization (ERS) occur, providing insights for BCI applications. MI-EEG-based BCIs are crucial in rehabilitation, enabling motor-impaired individuals to control assistive devices via neural signals, enhancing independence, and potentially inducing neuroplasticity. However, decoding MI–EEG signals is challenging due to their low signal-to-noise ratios [5], non-stationarity [6], and inter-individual variability. Overcoming these formidable obstacles is of paramount importance for actualizing the full potential of MI-EEG-based BCIs [7], thereby effectuating a paradigm shift in human–machine interaction and ameliorating the quality of life for those grappling with motor disabilities.

Although machine learning techniques are widely applied to decode Motor Imagery Electroencephalogram (MI-EEG) signals, their reliance on manual feature extraction prevents the discovery of deep latent features, thereby limiting decoding performance.To address these challenges, end-to-end deep learning approaches are increasingly being applied to EEG decoding. Unlike traditional machine learning methods that rely on manual feature extraction, end-to-end deep learning models can automatically learn and extract hierarchical features directly from raw EEG data. Many studies in recent years have applied deep learning methods such as convolutional neural networks (CNNs), long short-term memory networks (LSTMs) [8], and recurrent neural networks (RNNs) [9] to EEG signals, achieving good decoding results.

CNNs have demonstrated significant potential in EEG signal processing due to their ability to extract features automatically. However, CNNs primarily focus on local spatial patterns within the EEG signals and have limitations when it comes to dealing with the complex relationships between different EEG channels, which are crucial for understanding the brain’s neural activity patterns during motor imagery tasks [10]. Graph convolutional networks (GCNs) [11], on the other hand, are specifically designed to handle graph-structured data. In the context of EEG analysis, the EEG channels can be regarded as nodes of a graph, and the connections between channels can be represented by edges. GCNs can utilize the topological structure of the graph to accurately capture the channel-to-channel relationships [12], which is essential for a comprehensive understanding of MI-EEG signals.

Understanding the crucial importance of channel connectivity, temporal dependency, and frequency domain information in MI-EEG for enhancing decoding performance [13,14], we propose a novel dual-branch network architecture that integrates these dimensions for a comprehensive analysis. One branch uses Adaptive GCN to establish the connectivity between channels, capturing the synchronization state and dynamic process of the brain, and employs a Convolutional Block Attention Module (CBAM) to focus on important features, achieving spatial–spectral feature extraction. The other branch employs temporal convolution to serialize the MI-EEG, utilizing bidirectional gated recurrent unit (Bi-GRU) and multi-head self-attention mechanisms to extract deep temporal-spectral features from the dependency relationships in the sequence. Feature fusion is then performed. The main contributions of our work are as follows:An adaptive graph convolutional network is proposed, which constructs the graph convolutional layers of MI-EEG signals using a dynamic adjacency matrix. The CBAM is employed to focus on important features, capturing the synchronized activity states and dynamic processes of the brain, thereby improving the quality of spatial–temporal feature decoding.We propose a time–frequency feature extraction model to fully explore the sequential dependencies and global dependencies among features of different time segments. This study combines the Bi-GRU with the multi-head self-attention mechanism to obtain deep-level time–frequency features.Experiments on multiple public EEG datasets show that, compared with state-of-the-art methods, the proposed model achieves significant improvements, enhancing the decoding quality of MI-EEG multi-classification tasks.

The rest of this paper is organized as follows. In Section 2, some related works are first reviewed. Section 3 describes the proposed network structure. Section 4 details the experimental results. Section 5 discusses the effect of the proposed method on the utilization of channel correlation and temporal dependence. Finally, the paper is summarized in Section 6.

## 2. Related Works

### 2.1. Deep Learning Advancements

With the rapid development of deep learning technology, numerous recent studies have begun to apply deep learning methods to MI-EEG signal decoding and have achieved remarkable results. For instance, Alhagry et al. [15] proposed using the LSTM method to study the time-varying characteristics of EEG signals. Li et al. [16] proposed a bidirectional ConvLSTM model that combines the advantages of CNNs and LSTMs, enabling better exploration of deep temporal features in EEG signals and demonstrating enhanced modeling capabilities for time-series-related brain-electrical signals. Schirrmeister et al.’s [17] ShallowConvNet model and Lawhern V.J. et al.’s [18] EEGNet model were designed with optimizations tailored to the characteristics of EEG signals, achieving good classification results in an end-to-end manner without relying on prior knowledge. Additionally, models such as Multi-branch-3D [19] and MSFBCNN [20] introduced multi-branch structures and special convolution operations, further enhancing the classification performance of MI-EEG signals. These deep learning methods, based on convolution and recurrent units, can automatically learn hidden features within the signals through network training, avoiding the limitations of manual feature extraction and thus showing great potential in the field of MI-EEG signal decoding.

### 2.2. Graph Convolutional Networks and Feature Fusion

Despite the progress made by deep learning methods in MI-EEG signal decoding, several challenges remain. In particular, when dealing with EEG signals, due to the significant differences in activation intensities across different brain regions during motor imagery and the irregular spatial distribution of EEG signals, traditional CNNs struggle to effectively capture the connectivity between MI-EEG channels. To address this issue, graph convolutional networks (GCNs) have emerged. Recent studies have demonstrated that GCNs can model the relationships between EEG signal channels using adjacency matrices, thereby better extracting spatial information from the signals. For example, Zhang et al.’s [21] GCB-Net generalized graph convolutional network successfully extracted deeper spatial–spectral information by constructing an effective graph structure. Song et al.’s [22] dynamic graph convolutional neural network further enhanced the modeling of signal spatial features by dynamically learning the graph structure of EEG signals through network training. Chen et al. [23] combined the self-attention mechanism with spatial graph convolution, fully exploiting the correlations between EEG channels and constructing a classification model capable of obtaining richer spatial features.

In addition to spatial features, the channel connectivity, temporal dependency, and frequency domain information of MI-EEG signals are all vital for improving decoding performance. This is because the signals induced by motor imagery in the sensory-motor cortex exhibit ERD/ERS at specific rhythms, brain regions, and time segments. To fully utilize multi-domain features, some researchers have proposed feature fusion methods. Tang et al.’s [24] SF-TGCN model achieved deep temporal and spatial feature fusion of MI-EEG signals using multilayer temporal convolution modules and the Laplacian operator, making progress in capturing signal features in both the time and space dimensions. Li et al. [25] utilized temporal convolution blocks and multi-level wavelet convolution to achieve temporal-frequency feature fusion, achieving an accuracy of 74.71% in four-class motor imagery classification tasks. However, most existing studies only aggregate information from two dimensions and fail to fully utilize temporal, frequency, and spatial information simultaneously. This provides the research space and innovation opportunity for the present study to propose a method that can simultaneously explore the multi-domain features of MI-EEG signals.

## 3. Materials and Methods

### 3.1. Dataset

This study utilized the BCI-IV Dataset 2a and BCI-III Dataset 3a, which involve four categories of body movement imagery tasks: left hand, right hand, both feet, and tongue.

The BCI-IV Dataset 2a comprises EEG data from nine subjects (A01-A09), with each subject performing 576 motor imagery tasks. Each task lasted approximately 8 s, including a 2-s task warning period, a 4-s motor imagery period, and a 2-s rest period. EEG signals were recorded from twenty-two channels at a sampling frequency of 250 Hz. Post-acquisition, bandpass filtering from 0.5 to 100 Hz and 50 Hz notch filtering was applied (Figure 1).

The BCI-III Dataset 3a includes EEG data from three participants (K3b, K6b, L1b), with K3b having 360 trial sets and the other two participants having 240 trial sets each. Each imagery task lasted 4 s, with EEG signals recorded from 60 channels at a sampling frequency of 250 Hz. To align with the channel selection in BCI-IV Dataset 2a, 22 electrodes near the central scalp region were selected.

### 3.2. Proposed Model Framework

This work proposes a dual-branch network using Adaptive GCN and Bi-GRU to enhance MI-EEG decoding performance by simultaneously mining channel correlation and temporal dependency. The model architecture, shown in Figure 2, includes preprocessing, spatio-spectral feature extraction, temporal-spectral feature extraction, and feature fusion. In preprocessing, the original signal is segmented into nine sub-bands, with spatial filtering applied to each to obtain ERD/ERS modes across various frequency bands. Branch I utilizes Adaptive GCN and CBAM to extract spatial-spectral features from EEG channel functional connectivity, leveraging an adaptive adjacency matrix to capture synchronous brain activity states and dynamic processes. Branch II employs Bi-GRU and MHA to extract temporal-spectral features by capturing temporal dependencies in multi-band EEG sequences. Finally, a feature layer fusion strategy is adopted to integrate diverse information, which is then sent to the Softmax layer for classification.

### 3.3. Preprocessing

Distinguishing motor imagery actions can be achieved through event-related synchronization (ERS) and desynchronization (ERD) [26], which are prominent in the α (8∼13 Hz) and β (14∼30 Hz) bands. To preserve ERD/ERS modes at different frequencies, sub-band division and CSP-based spatial filtering are applied to the original MI-EEG, inspired by the FBCSP [27] algorithm.

A filter bank was constructed using nine overlapping Chebyshev Type II bandpass filters, each with a 6 Hz bandwidth, covering a range of 4∼42 Hz (4∼10, 8∼14, …, 36∼42 Hz). Each sub-band is spatially filtered as follows:(1)$$Z_{b,i}={\bar{W}}_{b}^{T}E_{b,i}$$
where $Z_{b,i}$ is the signal after spatial filtering, $E_{b,i}\in R^{c\times t}$ represents the *i*-th EEG sample in the *b*-th frequency band, *c* is the number of EEG channels, *t* denotes the number of sampling points per channel, and T is the transpose. ${\bar{W}}_{b}$ represents the CSP projection matrix derived from the eigenvalue decomposition of Equation (Equation 2): (2)$$\Sigma_{b,1}W_{b}=\left(\Sigma_{b,1}+\Sigma_{b,2}\right)W_{b}D_{b}$$
where $\Sigma_{b,1}$ and $\Sigma_{b,2}$ represent the covariance matrix estimates for the first and second classes of MI tasks in the b-th frequency band, respectively, $D_{b}$ is a diagonal array consisting of the eigenvalues of $\Sigma_{b,1}$. The first *n* and the last *n* column vectors from $W_{b}\in R^{c\times c}$ are selected to form the spatial domain filter ${\bar{W}}_{b}$. For the four-class MI tasks, a One-vs-Rest strategy is adopted. One MI task is considered the first class, while the other tasks are combined into the second class. The final spatial filter ${\bar{W}}_{b}$ contains k×2×n vectors, where k=4 represents the four classifications, and n=2 is set. After spatial filtering according to Equation (Equation 1), the number of EEG channels is 16. After spatial filtering of the 9 band signals, the dimension of the input data $f_{in}$ becomes 9×1000×16.

### 3.4. Spatio-Spectral Feature Extraction Module

EEG exhibits complex dynamic functional connections between different brain regions. Graph convolutional networks have been proven to capture the correlation between channels effectively. Therefore, a spatio-spectral feature extraction module is designed to capture EEG channel correlations across frequency bands and fully extract spatio-spectral features.

The topology of MI-EEG must first be constructed to process EEG signals using graph convolution [28]. An undirected EEG signal connectivity graph is denoted as G={V,E,A}, where $V=\{v_{1},v_{2},\ldots ,v_{i},\ldots v_{N}\}$ represents the set of nodes, $v_{i}$ denotes the electrode channel, E is the set of edges connecting the nodes, and (vi,vj)∈E, $A=\{a_{ij}\in R^{N\times N};i,j=1,\ldots ,N\}$ represents the adjacency matrix describing the connection strength between any two nodes. In constructing ***A***, the traditional approach considers the electrode distribution as a non-Euclidean space, representing the spatial disorder of the local EEG channel by the adjacency of electrode nodes. Since the ERD/ERS phenomenon in MI-EEG is distributed across multiple brain regions, electrode interactions must be reflected globally. In this paper, ***A*** is constructed using the Pearson correlation coefficient ***P***, as in Equation (Equation 3), which captures synchronous brain activity information and retains the negative correlation of EEG channel activity by taking the absolute value of ***P***. Additionally, introducing the self-connectivity coefficient α enhances the weight of the self-node.(3)A=∣P∣+αI

Since motor imagery is a dynamic process, Adaptive GCN [29] was used in constructing MI-EEG graphs to capture dynamically changing brain connectivity states. This method allows for adaptive adjustment of graph connection weights and enhances the extraction of deep spatio-spectral features.

Specifically, the spatio-spectral features ${\tilde{f}}_{sf}$ obtained by Adaptive GCN are computed as follows: (4)$${\tilde{f}}_{sf}=W_{1\times 1}*\left({\tilde{D}}^{-1/2}A{\tilde{D}}^{-1/2}+B+C\right)f_{in}$$

The adjacency matrix $A\in R^{N\times N}$ is calculated using Equation (Equation 3), and $\tilde{D}$ is the degree matrix used to standardize A. $B\in R^{N\times N}$ represents the shared adjacency matrix for all samples, retaining common information. It is a trainable parameter that is dynamically updated during model training. The update process follows Equation (Equation 5), with *r* as the learning rate and Loss as the model training loss.(5)$$B=\left(1-r\right)B+r\frac{\partial Loss}{\partial B}$$

$C\in R^{bs\times N\times N}$ represents the adjacency matrix associated with the sample data, assigning unique connection strengths to each data input, computed as in Equation (Equation 6), and $W_{\theta }$ and $W_{\phi }$ denote two 1 × 1 convolution kernels.(6)$$C=softmax(f_{in}^{T}W_{\theta }^{T}W_{\phi }f_{in})$$

Redundant information exists in the spatio-spectral features ${\tilde{f}}_{sf}$ extracted by Adaptive GCN (AGCN). Therefore, this paper introduces CBAM [30] (Figure 3), which uses an attention mechanism in both feature space and feature channel dimensions to suppress redundant features and improve the quality of the spatio-spectral features.

Feature channel attention emphasizes important feature channels. The spatial dimensionality of ${\tilde{f}}_{sf}$ is compressed using an average and maximum pooling to generate the matrices $F_{avg}^{c}$ and $F_{max}^{c}$, respectively. A multilayer perceptron (MLP) then generates the channel attention mapping $M_{c}$: (7)$$M_{c}\left(F\right)=\sigma \left(W_{1}\left(W_{0}\left(F_{avg}^{c}\right)\right)+W_{1}\left(W_{0}\left(F_{max}^{c}\right)\right)\right)$$
where σ represents the sigmoid function, and $W_{0}$ and $W_{1}$ are the weights of MLP. Feature space attention emphasizes the target region of interest, complementing channel attention. It is computed as in Equation (Equation 8) by applying average and maximum pooling along the feature channel dimensions to obtain the matrices $F_{avg}^{s}\in R^{1\times 1000\times 16}$ and $F_{max}^{s}\in R^{1\times 1000\times 16}$. These matrices are concatenated and then convolved with a 3×3 kernel to generate $M_{s}$.(8)$$\begin{array}{cc} M_{s}\left(F\right)\; & =\sigma (f^{3\times 3}\left(\left[AvgPool\left(f_{out}^{\prime }\right);MaxPool\left(f_{out}^{\prime }\right)\right]\right))\\ \; & =\sigma (f^{3\times 3}\left(\left[F_{a\nu g}^{s};F_{max}^{s}\right]\right)) \end{array}$$

The feature ${\tilde{f}}_{sf}$ is element-wise multiplied with $M_{c}\in R^{8\times 1\times 1}$ and $M_{s}\in R^{1\times 1000\times 16}$ to obtain the weighted spatio-spectral feature $f_{sf}$.(9)$$\begin{array}{c} f_{out}^{\prime }=M_{c}\left({\tilde{f}}_{sf}\right)⊗{\tilde{f}}_{sf}\\ f_{sf}=M_{s}\left(f_{out}^{\prime }\right)⊗f_{out}^{\prime } \end{array}$$

Figure 4 presents the implementation details of the AGCN layer, which combines the AGCN layer, batch normalization (BN), and the ReLU activation function to form a foundational graph convolutional network (GCN) module. Multiple GCN modules are then stacked to construct the complete graph convolutional network.

### 3.5. Temporal–Spectral Feature Extraction Module

MI-EEG is a highly time-varying signal, dependent on preceding and succeeding time segments [31]. To fully leverage the temporal dependencies and enhance MI-EEG’s decoding performance, this paper proposes a temporal-frequency feature extraction module, comprising a temporal convolution layer, a single-layer Bi-GRU, and an MHA module. The temporal convolution layer consists of a two-dimensional convolution filter of size (50, 1). The input is $f_{in}\in R^{9\times 1000\times 16}$, and the output is the temporal–spectral feature $x\in R^{16\times 191}=[x_{1},$…$,x_{t},$…$,x_{191}]$. Each sequence $x_{t}$ in x aggregates the temporal–spectral information of a 0.2 s time window (the sampling frequency is 250 Hz, and 50 sampling points are recorded in 0.2 s). The Bi-GRU [32] model captures the forward and backward dependencies of the sequence $x_{t}$. Bi-GRU, a type of RNN with a gating structure, includes an update gate $z_{t}$ and a reset gate $r_{t}$. This gating structure selectively transfers information in the hidden layer, solving the gradient vanishing problem in RNNs and overcoming short-term memory issues, as calculated by the following formulas:(10)$$r_{t}=\sigma \left(W_{r}x_{t}+U_{r}h_{t-1}+b_{r}\right)$$(11)$$z_{t}=\sigma \left(W_{z}x_{t}+U_{z}h_{t-1}+b_{z}\right)$$

W, U, and b are the weight parameter matrices and bias, respectively. $h_{t-1}$ represents the hidden state of the (t−1)th temporal–spectral sequence containing all information before the (t−1) time slice. σ is the sigmoid nonlinear activation function. Using the update gate $z_{t}$ and reset gate $r_{t}$, the hidden layer state information ${\vec{h}}_{t}$ of the current sequence $x_{t}$ is computed as shown in Equation (Equation 12); ${\tilde{h}}_{t}$ represents the candidate hidden state, and ⊙ denotes element-wise multiplication.(12)$$\begin{array}{cc} \; & {\tilde{h}}_{t}=\tanh\left(W_{h}x_{t}+U_{h}\left(r_{t}⊙h_{t-1}\right)+b_{h}\right)\\ \; & {\vec{h}}_{t}=\left(1-z_{t}\right)⊙h_{t-1}+z_{t}⊙{\tilde{h}}_{t} \end{array}$$

Bi-GRU computes the hidden states ${\vec{h}}_{t}$ and ${\overset{\leftarrow }{h}}_{t}$ from the forward and backward directions, and concatenates them to obtain the temporal–spectral feature ${\overset{↔}{h}}_{t}$, which contains both forward and backward dependencies. To improve the model’s capacity to capture long-range dependencies within the sequences, the MHA [33] is employed to capture global dependencies among ${\overset{↔}{h}}_{t}$. The MHA module performs self-attention computations with multiple attention heads and outputs the final temporal–spectral feature $f_{tf}$ following a linear transformation; $W_{O}$ represents the weight matrix.(13)$$f_{tf}=\left[{\mathrm{head}}_{1};\ldots ;{\mathrm{head}}_{i};\ldots ;{\mathrm{head}}_{h}\right]W_{O}$$(14)$${\mathrm{head}}_{i}=softmax\left(\frac{{\mathrm{QW}}_{Q}^{i}\cdot {\left({\mathrm{KW}}_{K}^{i}\right)}^{T}}{\sqrt{d_{k}}}\right){\mathrm{VW}}_{V}^{i}$$

Each attention head is computed as in Equation (Equation 14), Q, K, and V are the matrices obtained by linear transformation of ${\overset{↔}{h}}_{t}$, $W_{Q}^{i}$, $W_{K}^{i}$, and $W_{V}^{i}$ represent the trainable weight matrices, and $d_{k}=16$ is the feature dimension of the sequence ${\overset{↔}{h}}_{t}$.

### 3.6. Feature Fusion

The extracted spatio-spectral features $f_{sf}$ and temporal–spectral features $f_{tf}$ are fused. The features $f_{sf}$ and $f_{tf}$ are reduced to 8 and 16 dimensions using average aggregation. The spliced features are linearly transformed using a single-layer neural network, as shown in Equation (Equation 15), with W and b as the weights and bias. The features are converted into probabilities using the Softmax function in Equation (Equation 16) for normalization and dividing EEG into four categories.(15)$$x^{\prime }=W^{4\times 24}\left(\left[\mathrm{Avgagg}\left(f_{sf}\right),\mathrm{Avgagg}\left(f_{tf}\right)\right]\right)+b^{4\times 1}$$(16)$$x_{out}^{\prime }\left(i\right)=\frac{\exp(x^{\prime }\left(i\right))}{\sum_{k=1}^{4}\exp\left(x^{\prime }\left(k\right)\right)},i=1,\ldots ,4$$

## 4. Results

### 4.1. Software and Hardware Environment

The models were implemented in a Python 3.9 environment using the PyTorch 1.13.1 framework and executed on a GeForce 3050 GPU. Model parameter settings are detailed in Table 1. The final results were derived from the average of five separate experiments, using an 8:2 split ratio for training and test sets.

### 4.2. Classification Results

The classification performance of our proposed dual-branch network across subjects is summarized in Table 2, with results obtained by averaging five experiments conducted under identical conditions. On average, the model achieved an accuracy of 82.16% and a Kappa value of 0.761. Notably, accuracy exceeded 90% for three participants. However, subjects A02 and A06 showed significantly lower accuracies, below 70%, indicating poor classification. To investigate these disparities, we examined the signal characteristics of the subjects, focusing on event-related desynchronization/synchronization (ERD/ERS) phenomena [34,35].

Utilizing Pfurtscheller’s method [36], this study plotted the ERD/ERS characteristic curves for subjects A03, K3b, A02, and A06. The analysis employed Equation (Equation 17), where *A* is the signal energy during motor imagery and *R* is the baseline signal energy, measured during a reference period set from −3 to −2.5 s. This detailed approach underscores the significance of signal characteristics in enhancing classification accuracy and highlights potential avenues for improving model performance in subjects with initially low classification accuracies.(17)$$ERD/ERS=\frac{A-R}{R}\times 100\%.$$

Figure 5a,b illustrate the ERD/ERS characteristic curves for subjects A03 and K3b, respectively. During motor imagery, movements of the left hand, right hand, both feet, and tongue generally exhibit distinct ERD/ERS characteristics, resulting in better classification outcomes. Figure 5c presents the ERD/ERS characteristic curves for subject A02, where consistent trends and similar energy changes are observed in the C3, C4, and Cz channels during different imagined movements. Figure 5d displays the ERD/ERS characteristic curves for subject A06, where significant ERD is observed in the Cz channel across all the proposed method’s imagined tasks, with consistent trend patterns. These results indicate that subjects A02 and A06 did not elicit corresponding responses in the brain regions associated with different limb movements, lacking separability and resulting in poor classification outcomes.

Table 3 presents the decoding outcomes results of our proposed method and seven baseline methods on the two datasets. The proposed method achieved 80.38% accuracy and 0.737 Kappa on BCI-IV Dataset 2a, and 87.49% accuracy and 0.833 Kappa on BCI-III Dataset 3a, outperforming CSP [37], FBCSP [38], Deep ConvNet [17], Shallow ConvNet [17], and EEGNet. Furthermore, the proposed method achieved the highest results on both datasets compared to the recently proposed EEG-Conformer [39] and LightConvNet [31] models. This superior performance is attributed to the proposed method’s consideration of synchronized activities and dynamic signal changes during the MI-tasks, fully exploiting channel correlation and temporal dependence of the MI-EEG, resulting in better decoding outcomes.

This superior performance is attributed to the proposed method‘s consideration of synchronized activities and dynamic signal changes during the motor imagery, fully exploiting channel correlation and temporal dependence of the MI-EEG, resulting in better decoding outcomes.

### 4.3. Ablation Experiments

The proposed dual-branch network extracts spatio-spectral and temporal–spectral features of MI-EEG using Adaptive GCN and Bi-GRU, respectively, and performs feature fusion. We conducted comparative experiments to validate the effectiveness of multi-domain feature fusion.

Branch1: Only Adaptive GCN and CBAM are used to extract spatio-spectral features.

Branch2: Temporal–spectral features are extracted using Bi-GRU and MHA.

Our method: Combines temporal–spectral and spatial–spectral features using a dual-branch network.

The experimental results after the feature fusion of the two datasets are shown in Table 4. We present the precision and recall for each type of task to evaluate the model’s performance in recognizing various tasks. For the BCI-IV Dataset 2a, the precision and recall of each task are consistent across the three experiments. However, the classification performance of Branch1 and Branch2 is significantly lower than that of feature fusion (our method). For the BCI-III Dataset 3a, when only using Branch1 and Branch2 for classification, each task’s precision and recall distribution are uneven. For example, in Branch1, the left-hand task achieves a high precision but a low recall, indicating that the left-hand task is misclassified as other categories when only spatial–spectral features are extracted. In contrast, the classification results of each task after feature fusion (our method) are more balanced and significantly higher than those of Branch1 and Branch2. This indicates that the model’s generalization performance is better after feature fusion, proving the importance of feature fusion and the complementarity of different dimensional information.

From Table 4, it can also be seen that the spatial–spectral features of the two datasets are more distinguishable than the temporal–spectral features. This is because the neural activity caused by limb movements has a distinct somatotopic organization. For example, hand movements mainly manifest in the contralateral brain hand functional area, and foot movements activate the central brain area. Compared to time-dimensional information, the spatial distribution differences of signals generated by different limb movements are more significant.

### 4.4. Comparison Experiments of Adjacency Matrices

The adaptive adjacency matrix in this study comprises three matrices: ***A***, ***B***, and ***C***, capturing synchronous brain activity and dynamic connections. Three experiments were conducted to verify their validity.

Experiment 1: Constructing the adjacency matrix using the spatial positioning information of EEG channels [40].

Experiment 2: Constructing the adjacency matrix using only matrix ***A***.

Experiment 3: Constructing the adaptive adjacency matrix using Equation (Equation 4) (our method). The experimental results of classification using different adjacency matrices for the two datasets are shown in Table 4. Comparing Experiment 1 and Experiment 2, adopting matrix ***A*** yielded better results than constructing the adjacency matrix based on spatial location information. This is because adjacency relationships can only aggregate information between channels in local brain regions, while synchronous brain activities occur in the whole brain region. Adjacency matrix ***A*** can establish connections across all EEG channels, making better use of channel information.

In Experiment 3, for the BCI-IV Dataset 2a, using the adaptive adjacency matrix (our method) improves the precision of each task by 6.99%, 5.49%, 7.12%, and 5.76%, and the recall by 7.58%, 6.13%, 7.82%, and 5.59% compared to Experiment 2. For the BCI-III Dataset 3a, using the adaptive adjacency matrix (our method) improves the precision of each task by 11.71%, 5.47%, 6.81%, and 1.33%, and the recall by 4.63%, 5.55%, 11.29%, and 13.52% compared to Experiment 2. The improvement is attributed to our method’s ability to establish whole-brain connectivity and adaptively capture the brain’s dynamic processes, thereby fully exploring brain functional activities.

### 4.5. Effect of Self-Connection Coefficients

Introducing the self-connection coefficient α aims to enhance the utilization of self-channel information, with its value ranging from 0 to 1. To verify the impact of the self-connection coefficient on classification results, we tested values at intervals of 0.1. As shown in Figure 6, classification accuracy increases with the self-connection coefficient for most subjects, reaching a peak before decreasing. This indicates that placing too much or too little emphasis on the node’s information can negatively affect classification performance. The optimal self-connection coefficient varies among subjects, with values of 0.3, 0.1, 0.2, 0.7, 0.5, 0.1, 0.6, 0.6, 0.6, 0.2, 0.4, and 0.2, reflecting the individual differences among the subjects.

## 5. Discussion

We propose a dual-branch Ml-EEG decoding method using adaptive GCN and Bi-GRU, which can capture the channel correlations and temporal dependencies among multiple frequency bands. To better understand the advanced nature of the proposed method in capturing brain functional activities and temporal dependencies, we conducted a visual analysis of the adjacency matrix and the output characteristics of each stage of the model.

### 5.1. Channel Correlation Visual Analysis

The weight edges of the adjacency matrix ***A*** were visualized to evaluate the impact of graph convolution on MI-EEG channel correlation. Figure 7 shows the brain’s functional connectivity patterns of different subjects. The outer circle represents different EEG channels, and the inner lines indicate their connectivity properties. Darker lines show a greater correlation, based on Pearson correlation coefficients in adjacency matrix ***A***. Weights are assigned to channel pairs based on correlation, selectively aggregating synchronous activity information. Connection strength varies among subjects, indicating individual differences.

### 5.2. Time Dependence Visual Analysis

To observe the impact of temporal dependencies on features, the features x before capturing temporal dependencies and features $f_{tf}$ after capturing them for all samples of each subject were obtained, and the feature mean was calculated along the sample dimensions for visualization. The visualization for subjects A03 and K3b is shown in Figure 8. The horizontal axis displays sequences from various time windows, each aggregating data from 50 sample points (0.2 s time window), and the vertical axis represents the sequence features.

Before temporal dependency capture, the feature map is discrete, indicating weak connections between time window sequences. After mining, dependencies strengthen significantly. For example, feature channels 3 and 12 of subject A03 are extremely similar in color throughout the entire time axis, suggesting that these time segments aggregate each other’s information. This shows that Bi-GRU and MHA effectively extract MI-EEG’s temporal dependence.

In the feature map before temporal-dependency capture, the first 25 sequences show obvious differences and prominent colors, while other sequences have small feature values. This is because the first 25 sequences correspond to the initial 125 MI-EEG sampling points (0∼0.5 s period of the acquisition paradigm), which is the task queuing stage when subjects are in a preparation state. The data from this period generally lack useful information. After temporal-dependent mining, these features are no longer prominent, indicating that Bi-GRU and MHA effectively reduce irrelevant information.

### 5.3. Feature Fusion Visual Analysis

To visually demonstrate the classification effect of feature fusion, the t-SNE [41] method was used to embed the temporal–spectral features, spatial–spectral features, and fusion features (our method) into two-dimensional scatter plots to observe the degree of feature separation. Figure 9a shows the visualization results of subject A07. For temporal–spectral features, different tasks overlap significantly. For spatial–spectral features, the separability between left- and right-hand tasks and between feet and tongue tasks is poor. The inter-class distances between tasks are increased after feature fusion significantly. Figure 9b shows the visualization results of subject K3b. From the temporal–spectral and spatial–spectral features, it can be seen that the left- and right-hand tasks introduce other individual samples. However, after feature fusion, the left- and right-hand tasks are completely separated from the other tasks. This result indicates that integrating the time, frequency, and spatial domain information of MI-EEG can enhance the separability of different tasks.

### 5.4. Future Directions

The present study acknowledges certain limitations, notably that the model’s performance was evaluated solely on two small-sample datasets. This methodological decision was informed by several factors. First, both the BCI-IV Dataset 2a and the BCI-III Dataset 3a are well-established benchmark datasets within the EEG research community; their utilization facilitates direct comparisons with existing studies and enhances the credibility and reproducibility of our findings. Second, the two-stream network architecture proposed in this paper deliberately increases model complexity to achieve superior feature extraction capabilities. However, this enhanced complexity has the adverse effect of reducing the model’s generalizability. To address these limitations, future research will explore advanced techniques such as meta-learning and transfer learning. These strategies are expected to bolster the model’s robustness and performance in scenarios characterized by limited and highly variable data, ultimately improving its generalization across diverse datasets.

## 6. Conclusions

Aiming at the problems that the channel correlations and temporal dependencies among multiple frequency bands of MI-EEG are difficult to capture precisely and efficiently, and that the signal decoding quality needs to be improved, we propose a multi-domain feature classification method for MI-EEG based on a dual-branch network. This method uses Adaptive GCN and Bi-GRU to mine the brain functional connectivity and temporal dependencies of MI-EEG respectively. Meanwhile, the CBAM attention mechanism and multi-head self-attention mechanism are introduced to capture the valuable information in features of each dimension, achieving an effective fusion of time–frequency–spatial domain features of MI-EEG and improving the decoding quality. The proposed method is evaluated with two datasets, BCI—IV Dataset 2a and BCI—III Dataset 3a. The results show that our method has a higher performance and provides a new approach to MI-EEG decoding.

## Figures and Tables

**Figure 1 sensors-25-01147-f001:**
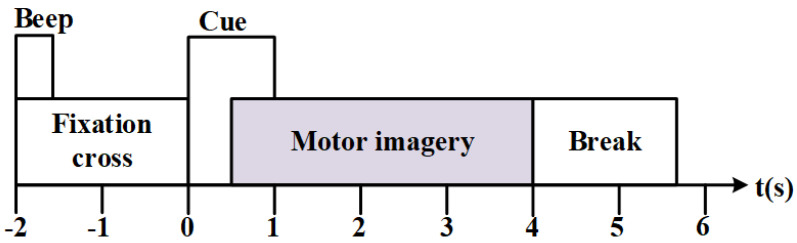
Schematic of CBAM.

**Figure 2 sensors-25-01147-f002:**
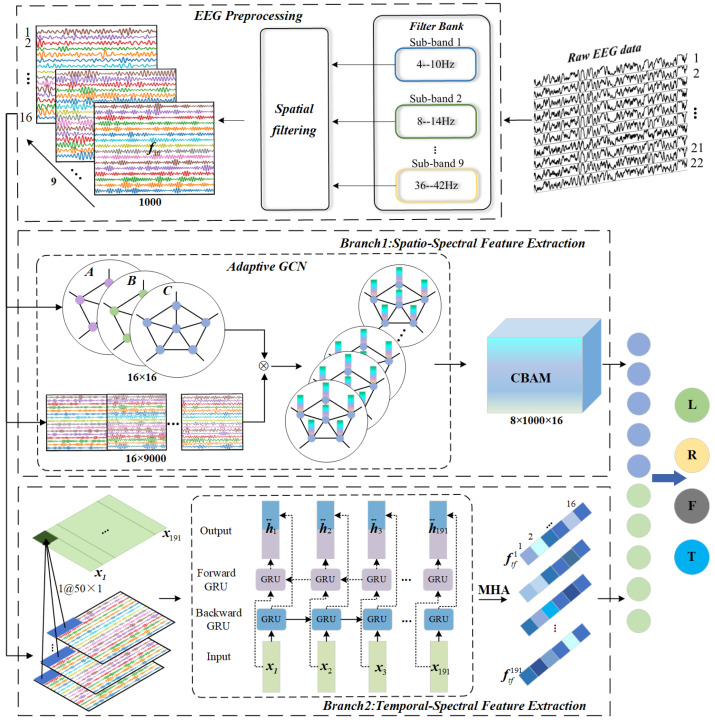
Implementation structure of a dual-branch network. A denotes the adjacency matrix between electrodes, B denotes the shared adjacency matrix between all samples, and C denotes the strength of the connection private to each sample.

**Figure 3 sensors-25-01147-f003:**
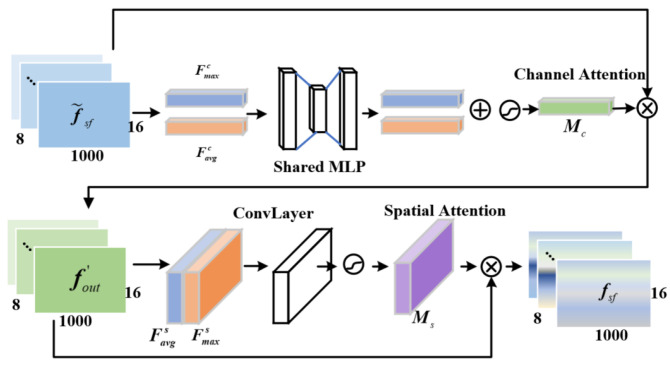
Schematic of CBAM.

**Figure 4 sensors-25-01147-f004:**
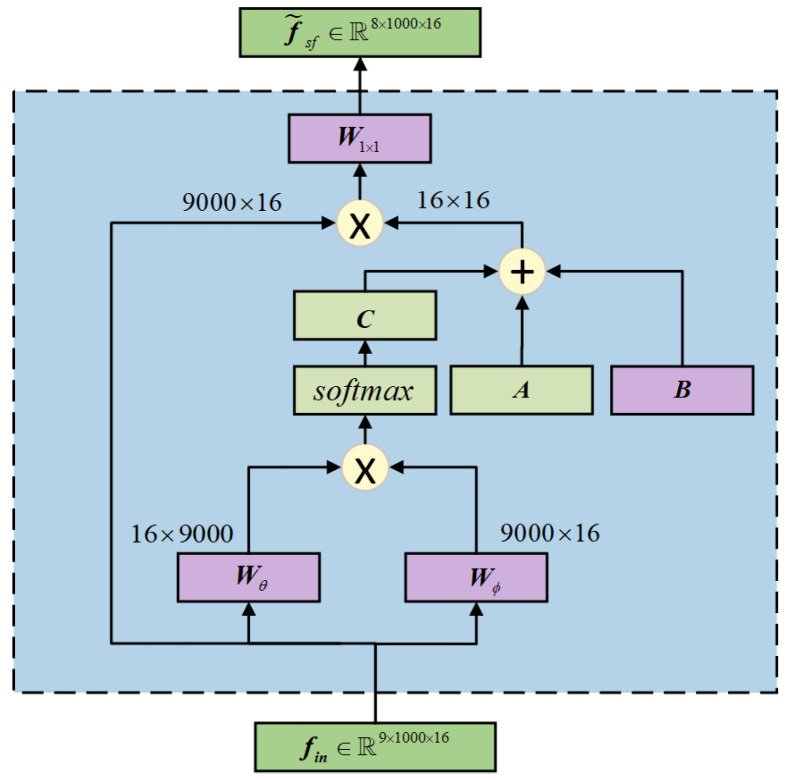
Schematic of the AGCN layer. ***A***, ***B***, ***C*** together form the adaptive adjacency matrix.

**Figure 5 sensors-25-01147-f005:**
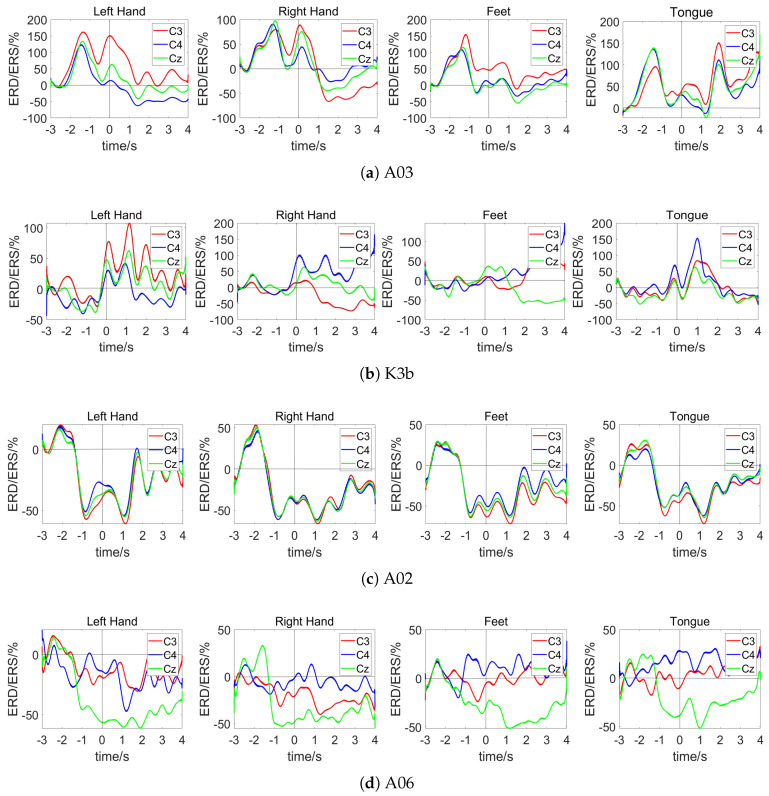
ERD/ERS characteristic curve.

**Figure 6 sensors-25-01147-f006:**
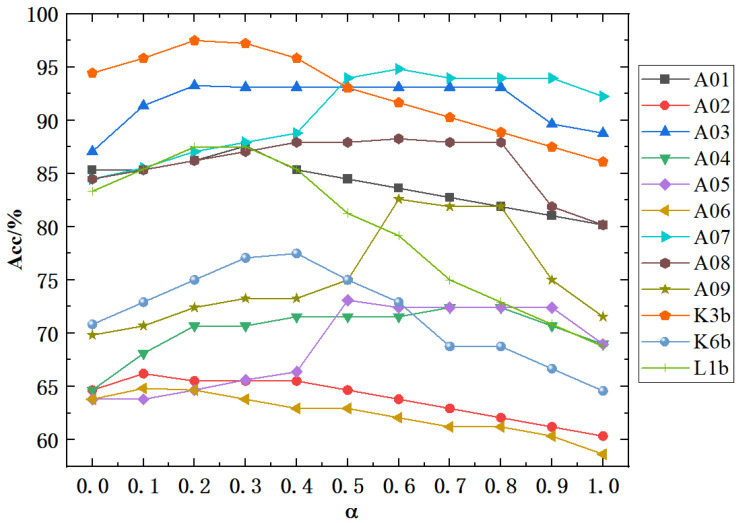
Visualization of adjacency matrix with different self-connection coefficients.

**Figure 7 sensors-25-01147-f007:**
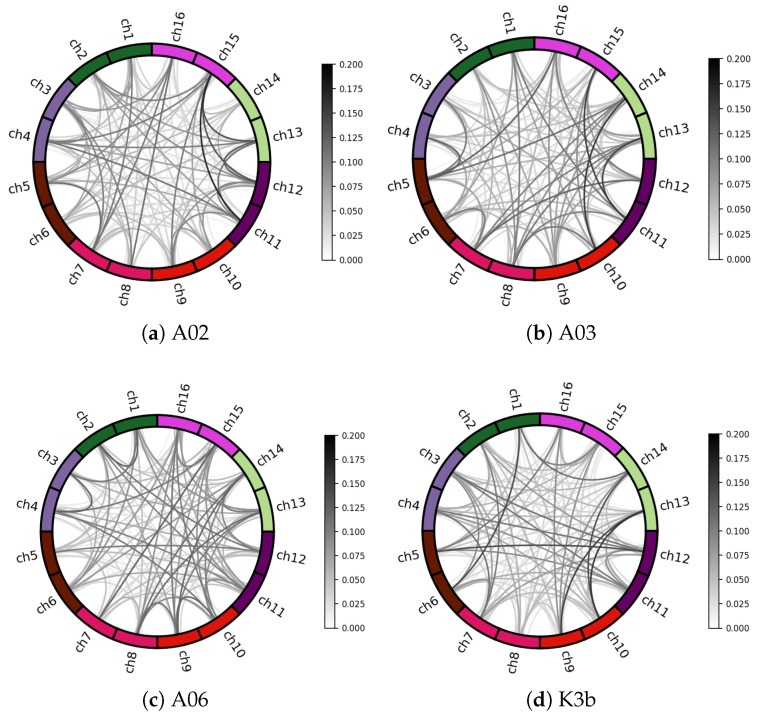
Spatial correlation connectivity maps. The outer circle is different EEG channels, the inner circle lines indicate the connectivity properties of different channels, and the darker color of the lines represents the greater correlation between the channels.

**Figure 8 sensors-25-01147-f008:**
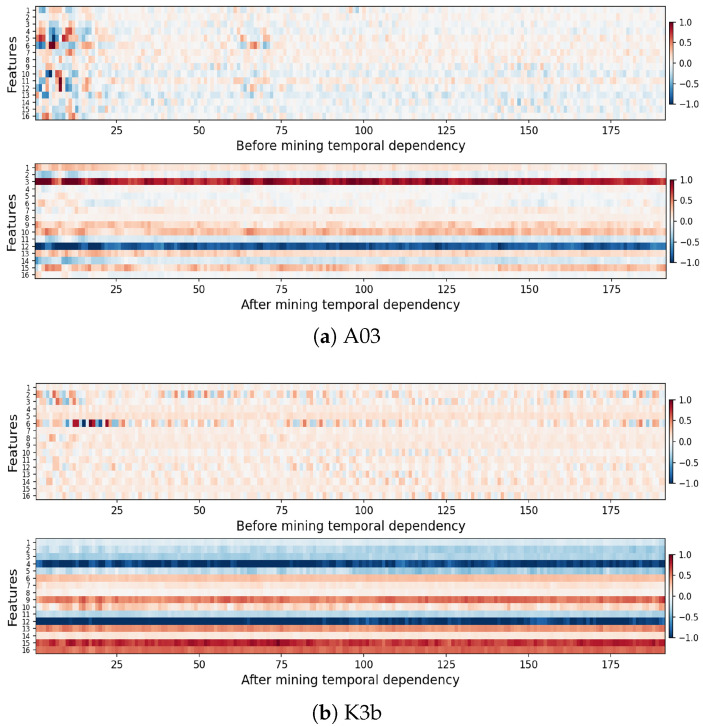
Feature visualization before and after temporal-dependent mining.

**Figure 9 sensors-25-01147-f009:**
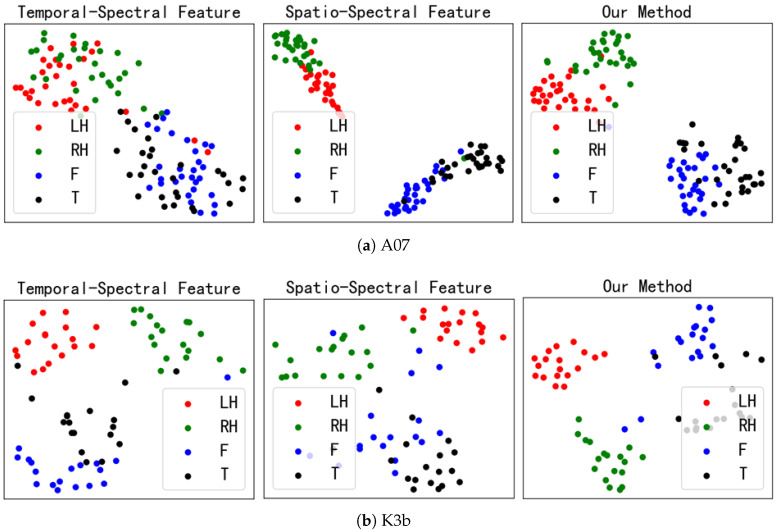
Visualization results for different features. The red circle is the left-hand MI task, the green circle is the right-hand MI task, the blue circle is the feet MI task, and the black circle is the tongue MI task.

**Table 1 sensors-25-01147-t001:** Parameter settings for the model.

Parameters	Setting
Optimizer	Adam
Loss function	Categorical cross-entropy
Learning rate	0.01
Batch size	16
Epochs	100

**Table 2 sensors-25-01147-t002:** Classification results of all subjects.

Subject	Acc%	Kappa	Precision %					Recall %			
			Left	Right	Feet	Tongue		Left	Right	Feet	Tongue
A01	88.18	0.834	94.26	90.37	84.59	83.47		86.89	93.79	84.14	85.51
A02	66.21	0.549	57.57	53.04	87.25	68.57		60.00	51.72	84.13	68.96
A03	93.26	0.910	90.67	99.31	92.99	91.20		97.93	95.86	88.96	90.34
A04	72.41	0.632	69.61	70.34	77.41	74.36		68.96	70.34	82.07	68.27
A05	73.09	0.641	83.38	75.87	64.32	71.01		71.03	88.96	61.38	71.03
A06	64.65	0.528	64.90	60.42	68.69	67.11		66.21	59.99	65.51	66.90
A07	94.82	0.931	95.20	95.92	94.03	94.51		93.79	96.55	95.17	93.79
A08	88.27	0.843	87.96	84.99	88.94	92.62		94.48	91.72	81.38	85.51
A09	82.58	0.767	78.92	85.84	80.62	87.54		89.65	71.72	78.62	90.34
K3b	97.49	0.966	100.0	96.84	94.84	98.94		97.77	100.0	98.88	93.33
K6b	77.49	0.699	75.34	70.34	75.12	90.86		63.33	71.66	85.00	90.00
L1b	87.49	0.833	91.60	90.86	81.98	70.09		86.66	95.00	81.66	86.66
Average	82.16	0.761	82.45	81.18	82.56	82.52		81.39	82.27	82.24	82.55

**Table 3 sensors-25-01147-t003:** Parameter settings for the model.

Dataset	Method	Acc%	Kappa
BCI-IV Dataset 2a	CSP+SVM	63.22 ± 17.08	0.508 ± 0.230
	FBCSP+SVM	69.25 ± 15.49	0.589 ± 0.206
	Shallow ConvNet	79.21 ± 11.87	0.722 ± 0.158
	Deep ConvNet	75.65 ± 14.58	0.675 ± 0.194
	EEGNet	70.49 ± 15.68	0.606 ± 0.209
	EEG-Conformer	79.18 ± 9.625	0.722 ± 0.128
	LightConvNet	74.54 ± 12.20	0.659 ± 0.164
	**Our method**	**80.38** ± **10.89**	**0.737** ± **0.144**
BCI-III Dataset 3a	CSP+SVM	69.90 ± 16.86	0.598 ± 0.224
	FBCSP+SVM	75.69 ± 14.66	0.675 ± 0.195
	Shallow ConvNet	80.08 ± 14.39	0.734 ± 0.191
	Deep ConvNet	81.57 ± 8.614	0.754 ± 0.114
	EEGNet	76.84 ± 14.26	0.691 ± 0.190
	EEG-Conformer	81.93 ± 10.03	0.759 ± 0.133
	LightConvNet	87.36 ± 0.783	0.831 ± 0.104
	**Our method**	**87.49** ± **8.164**	**0.833** ± **0.108**

**Table 4 sensors-25-01147-t004:** Classification results of all subjects.

Dataset	Method	Precision %					Recall %			
		Left	Right	Feet	Tongue		Left	Right	Feet	Tongue
Dataset 2a	Branch1	73.26 ± 15.71	72.38 ± 17.68	72.19 ± 12.54	76.95 ± 12.34		72.48 ± 17.72	71.64 ± 19.20	73.33 ± 14.78	73.56 ± 12.27
	Branch2	71.32 ± 16.60	71.62 ± 14.10	67.88 ± 10.90	72.25 ± 13.27		70.88 ± 16.88	70.72 ± 20.62	69.23 ± 10.90	68.73 ± 10.55
	**Ours**	**80.27** ± **12.74**	**79.57** ± **14.95**	**82.09** ± **9.77**	**81.15** ± **10.33**		**80.99** ± **13.52**	**80.07** ± **15.96**	**80.15** ± **10.05**	**80.07** ± **10.41**
Dataset 3a	Branch1	88.42 ± 7.45	71.81 ± 14.32	74.21 ± 10.21	78.69 ± 14.22		69.44 ± 29.57	83.88 ± 21.67	76.10 ± 15.29	81.84 ± 8.121
	Branch2	72.81 ± 19.62	78.09 ± 16.64	77.96 ± 8.830	81.10 ± 9.475		75.55 ± 23.14	77.22 ± 20.65	68.51 ± 13.83	82.59 ± 4.286
	**Ours**	**88.98** ± **10.23**	**86.01** ± **11.34**	**83.98** ± **8.17**	**86.63** ± **12.15**		**82.59** ± **14.35**	**88.88** ± **12.34**	**88.51** ± **7.458**	**90.00** ± **2.72**

## Data Availability

Publicly available datasets were analyzed in this study. The BCI Competition IV Dataset IIa can be found here: https://www.bbci.de/competition/iv/(accessed on 5 November 2023). The BCI Competition III Dataset IIIa can be found here: https://www.bbci.de/competition/iii/ (accessed on 5 November 2023).

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
