# Peer review of "Adaptive GCN and Bi-GRU-Based Dual Branch for Motor Imagery EEG Decoding"

_sensors, 2025, doi:10.3390/s25041147_

Round 1

Reviewer 1 Report

Comments and Suggestions for Authors

- The manuscript lacks the comparative study. The authors need to compare to the previous proposed computational methods and their results/performance. 

- The computational experiments on both datasets (BCI-IV Dataset 2a and BCI-III Dataset 3a) combined together would be interesting.  

- In addition, the authors should consider to apply to other common datasets. 

- The authors need to make it clear that the computational experiments were randomized and run multiple times (this means that the results reported in the manuscript were not obtained from just a single computational experiment). 

Reviewer 2 Report

Comments and Suggestions for Authors

Dear authors my comments for this paper as follows: -

1.     Avoid citing with more than one or two references for one paragraph such as “Motor Imagery Brain-Computer Interface (MI-BCI) enables direct interaction be tween the brain and external devices by decoding electrical activity from the central nervous system[1][2][3].”

2.     The introduction section needs to be separated into  two parts namely, the introduction and the related work. Because the existing introduction section is full of related work and lack in discussing the research problem as well as the research gap.

3.     Compare with UP TO DATE  studies for BCI competition IV  dataset 2a.

4.     Compare with UP TO DATE  studies for BCI competition dataset 3a.

5.     NO need to discuss about machine learning and give a literature about related work , because current study uses deep learning example Machine learning techniques have been widely applied in the decoding of Motor Imagery Electroencephalogram (Ml-EEG) signals.The Common Spatial Pattern (CSP) is a commonly used method for spatial feature extraction[9], which enhances the separability of signals by maximizing the variance between classes.”

6.     Justify why using these BCI competition datasets , and why four class not two class?

7.     Justify why using dual-branch network using Adaptive GCN and Bi-GRU?

8.     Justify why using four second not two second from the four second motor imagery signal?

regards
